# Impacts of Emergency Medical Technician Configurations on Outcomes of Patients with Out-Of-Hospital Cardiac Arrest

**DOI:** 10.3390/ijerph17061930

**Published:** 2020-03-16

**Authors:** Pin-Hui Fang, Yu-Yuan Lin, Chien-Hsin Lu, Ching-Chi Lee, Chih-Hao Lin

**Affiliations:** 1Department of Emergency Medicine, National Cheng Kung University Hospital, College of Medicine, National Cheng Kung University, Tainan 70403, Taiwan; fph2005er@gmail.com (P.-H.F.); yuyuan1122@hotmail.com (Y.-Y.L.); jazojazojazo@gmail.com (C.-H.L.); 2Department of Internal Medicine, National Cheng Kung University Hospital, College of Medicine, National Cheng Kung University, Tainan 70403, Taiwan; 3Graduate Institute of Medical Sciences, College of Health Sciences, Chang Jung Christian University, Tainan 71101, Taiwan; 4Department of Adult Critical Care Medicine, Tainan Sin-Lau Hospital, Tainan 70142, Taiwan

**Keywords:** emergency medical technician, paramedic, out-of-hospital cardiac arrest, return of spontaneous circulation

## Abstract

Paramedics can provide advanced life support (ALS) for patients with out-of-hospital cardiac arrest (OHCA). However, the impact of emergency medical technician (EMT) configuration on their outcomes remains debated. A three-year cohort study consisted of non-traumatic OHCA adults transported by ALS teams was retrospectively conducted in Tainan City using an Utstein-style population database. The EMT-paramedic (EMT-P) ratio was defined as the EMT-P proportion out of all on-scene EMTs. Among the 1357 eligible cases, the median (interquartile range) number of on-scene EMTs and the EMT-P ratio were 2 (2–2) persons and 50% (50–100%), respectively. The multivariate analysis identified five independent predictors of sustained return of spontaneous circulation (ROSC): younger adults, witnessed cardiac arrest, prehospital ROSC, prehospital defibrillation, and comorbid diabetes mellitus. After adjustment, every 10% increase in the EMT-P ratio was on average associated with an 8% increased chance (adjusted odds ratio [aOR], 1.08; *p* < 0.01) of sustained ROSC and a 12% increase change (aOR, 1.12; *p* = 0.048) of favorable neurologic status at discharge. However, increased number of on-scene EMTs was not linked to better outcomes. For nontraumatic OHCA adults, an increase in the on-scene EMT-P ratio resulted in a higher proportion of improved patient outcomes.

## 1. Background

Out-of-hospital cardiac arrest (OHCA) is a global problem and is linked to high mortality and morbidity. The resuscitation science has significantly progressed, but survival from OHCA has not significantly improved in the past decades [1]. Annually, the emergency medical service (EMS) team treats approximately 176,100 OHCA in the United States [2]. In Taiwan, the incidence of OHCA was reported to be an average of 9815 cases per year and the 30-day and 180-day mortality rates were 75.8% and 86.0%, respectively [3]. Prehospital basic life support (BLS), consisting of recognizing cardiac arrest, activating the emergency response system, early cardiopulmonary resuscitation (CPR) and rapid defibrillation, is fundamental to OHCA resuscitation [4]. Advanced life support (ALS) is composed of more advanced management than BLS care, such as intravascular therapy and endotracheal intubation [5]. It remains unclear whether prehospital ALS provides more benefit to OHCA patients than BLS care. A previous meta-analysis has shown that implementation of ALS had a better survival rate at hospital discharge than BLS care in nontraumatic OHCA patients [6], whereas some recent observational studies suggested no benefit or even a reduction in survival rate from prehospital ALS [7,8,9]. 

On-scene emergency medical technicians (EMTs) are the first-line health care providers for OHCA patients worldwide. The training hours for EMT-paramedic (EMT-P) accreditation in Taiwan, which is 1280 h, is more than 4-fold that for EMT-intermediate (EMT-I) accreditation [10,11]. The EMT-P is capable of endotracheal intubation, defibrillation, intravascular therapy and transcutaneous pacing, which EMT-I cannot perform [12]. Since the level of EMTs differs in the resuscitation they can provide, the number and level of EMTs in the field are considered crucial determinants for the outcomes of OHCA patients. However, the impact of the EMT configuration on patient survival remains debated [13,14,15,16,17].

Numerous previous reports only discussed EMT numbers [13,14,16], whereas other investigations took both total EMT number and the contribution of EMT-Ps as exposure variables [15,17]. The EMT-P is capable of administering more advanced management [12] and is more confident in performing clinical activities and tasks [18]. Therefore, our study aims to investigate the benefit of the numbers and configurations of on-scene EMTs, particularly focusing on the impact of the EMT-P ratio, on the outcomes of OHCA patients.

## 2. Materials and Methods 

### 2.1. Study Design

A retrospective cohort study was conducted with the data of an Utstein-style population database from 1 January 2014 to 31 December 2016, in Tainan city, Taiwan. During the study period, nontraumatic OHCA adults (age ≥ 18 years) who activated the EMS teams and were treated by ALS teams were included. Cardiac arrest defined as the absence of signs of circulation, was confirmed by EMTs on the scene. Patients with a known pregnancy who were less than eighteen years old and had obvious signs of irreversible death, severe hypothermia, or valid do-not-attempt-resuscitate (DNAR) orders were excluded. The causes of cardiac arrest were classified as traumatic or nontraumatic based on the clinical judgments of the EMS providers and physicians in charge. A traumatic cardiac arrest was defined as a cardiac arrest resulting from a traumatic event; those not classified as traumatic were defined as nontraumatic. 

### 2.2. EMS in Tainan City

Tainan city is 2192 km^2^ in area and has 1.9 million people. There are two designated medical centers and 11 designated local hospitals in Tainan city. The EMS system is fire-bureau-based, with one centralized dispatch center. The EMS dispatch center is operated by experienced EMTs or registered nurses 24 h a day. The dispatchers would dispatch ALS teams should events of cardiac arrest be recognized in the emergency calls. Otherwise, the dispatchers usually dispatch the nearest teams to the scene. EMTs were classified into three levels: EMT-basic, EMT-I, and EMT-P. There are 52 EMS stations and 615 EMTs in Tainan city, which include 5 EMT-basics, 493 EMT-Is, and 117 EMT-Ps. The average age of EMTs is 37 years and 580 (94%) of them are male. 

During the study period, all EMTs performed CPR according to the Taiwan Guidelines based on the American Heart Association, European Resuscitation Council, and the International Liaison Committee on Resuscitation 2010 Guidelines [19]. For patients with OHCA, CPR efforts performed by an EMT-basic may include BLS with defibrillation and the use of bag-valve-masks for ventilation, while the efforts performed by an EMT-I may include BLS with defibrillation and placement of advanced airways such as laryngeal mask airways. The resuscitation efforts performed by an EMT-P could include defibrillation with ALS, such as endotracheal intubation and intravenous administration of epinephrine, as per protocol [20]. EMT-Ps may use a laryngeal mask airway instead of endotracheal intubation since endotracheal intubation did not necessarily provide better outcomes for patients with OHCA [21].

Use of automated external defibrillators was mandated during the resuscitation for nontraumatic OHCA. Patients with nontraumatic OHCA received CPR for at least two full cycles (approximately 5 min) before being transported to a designated hospital. Rules for the termination of resuscitation in prehospital settings did not exist, so resuscitation efforts continued during transportation to a hospital unless return of spontaneous circulation (ROSC) was achieved.

### 2.3. Definition of Crew Number, ALS Team, and EMT-Paramedic Ratio

The number in a crew was defined as the number of EMTs present in one ambulance run. The crew in one ambulance run typically includes 2 to 4 EMTs, depending on the availability of human resources. An ALS team was defined as a team that included at least one EMT-P. The EMT-P ratio was defined as the EMT-P proportion out of all on-scene EMTs, that is, the number of EMT-Ps divided by the crew total in one ambulance run. 

### 2.4. Exposure and Outcome

Data were obtained from a prospective registry database using computer interfaces. The collected data included the information required for the international Utstein-style criteria, including the patient demographics, witness status, past medical history, EMS response time (defined as the time from the ambulance departure from the fire station to the ambulance arrival at the scene), EMS time at the scene (defined as the time from the ambulance arrival at the scene to the ambulance departure from the scene), EMS transport time (defined as the time from the ambulance departure from the scene to the ambulance arrival at the hospital), initial cardiac rhythms, presence of bystander CPR, extent and amount of prehospital emergency care, achievement of ROSC at any time, a sustained (≥2 h) ROSC, survival at discharge and the neurologic status at discharge.

The exposures in our study were defined as the EMT configurations, including the total crew number and the EMT-P ratio. As previously described [22], the primary outcome was the achievement of a sustained (≥2 h) ROSC; and the secondary outcomes were any ROSC, survival at hospital discharge, and favorable neurologic status at discharge, in terms of the cerebral performance category (CPC) level I and II. 

### 2.5. Statistical Analysis

Statistical analyses were performed using the Statistical Package for the Social Sciences for Windows (Version 20.0, Chicago, IL, USA). Since the crew number in one ambulance run in Tainan city was generally between 2 and 4, the EMT-P ratio was assessed as a continuous variable and was stratified into four categories: 25.0–33.3%, 50%, 66.7–75.0%, and 100%. The correlation between the EMT-P ratio and other variables was analyzed by Pearson’s correlation. To recognize the independent predictors of sustained ROSC, all variables with *p* values less than 0.1 by bivariate analysis and those proven in previous investigations were processed for the stepwise backward logistic regression model. After adjusting all independent predictors of sustained ROSC, the beneficial influence of the average increase in the EMT number and the EMT-P ratio on patient outcomes was studied using logistic regression. A *p* value less than 0.05 was considered statistically significant.

We did not conduct formal sample size calculations, and all available data were used, to maximize the power. At least 8 to 10 events per variable are needed for a reliable multiple logistic regression analysis [23]. As for missing values, we conducted a complete case analysis if the missing values made up less than 5% of the values; in this case, an analysis might be feasible. If the missing values were at or above 5%, we performed appropriate imputation [24]. 

### 2.6. Ethical Consideration

The study was in accordance with ethical standards and was approved by the Institutional Review Board in National Cheng Kung University Hospital (A-ER-105-363).

## 3. Results

### 3.1. Patient Population

During the study period, 8150 patients with OHCA activated the Tainan EMS system. A total of 4498 adults with non-traumatic OHCA were considered after excluding 83 patients below 18 years old, 966 patients with traumatic OHCA, and 2603 patients with no resuscitation attempts because of valid DNR orders or other reasons. Among them, 1357 (30.5%) patients who were treated by ALS teams were eligible for the analysis. Figure 1 provides an overview of the OHCA case evaluation. 

The mean (standard deviation, SD) age of enrollees was 68.8 (16.4) years and 884 (65.1%) of them were male. The mean EMS response time, EMS scene time, and EMS transport time were 6.3, 13.2, and 5.0 min, respectively. The median (interquartile range, range) number of on-scene EMTs and the EMT-P ratio were 2 (2–2, 2–4) persons and 50% (50–100%, 25–100%), respectively. The patients were stratified into 4 EMT-P ratio groups 25–33.3%, 40–50%, 66.7–75%, and 100%. 

Table 1 reveals the trends of patient demographics, EMS time interval and prehospital treatment as the EMT-P ratio increased. There were no significant differences among the 4 groups regarding patient characteristics, witness of cardiac arrest, bystander CPR, or prehospital management, such as laryngeal mask airway use, defibrillation, and intravenous epinephrine (all *p* ≥ 0.05). A negative ratio-related trend in EMS transport time of ≤5 min (*γ* = −1.00, *p* = 0.01) and a positive ratio-related trend in comorbid hemato-oncological disease (*γ* = 1.00, *p* = 0.01) were seen. A positive ratio-related trend in the achievement of a sustained ROSC (*γ* = 1.00, *p* = 0.01) was exhibited (Figure 2).

### 3.2. Clinical Predictors of Sustained ROSC

Statistically significant associations of several variables with sustained ROSC, including younger adults (odds ratio [OR] 1.75, 95% confidence interval [CI] 1.39–2.27, *p* < 0.01), witnessed cardiac arrest (OR 3.31, 95% CI 2.55–4.30, *p* < 0.01), EMS response time ≤ 5 min (OR 0.76, 95% CI 0.59–0.97, *p* = 0.03), indoor location of cardiac arrest (OR 0.57, 95% CI 0.39–0.82, *p* < 0.01), prehospital ROSC (OR 17.13, 95% CI 9.82–29.87, *p* < 0.01), prehospital defibrillation (OR 2.33, 95% CI 1.73–3.13, *p* < 0.01), prehospital use of intravenous epinephrine (OR 1.44, 95% CI 1.03–2.01, *p* = 0.03), and comorbidities of diabetes mellitus (OR 1.37, 95% CI 1.05–1.78, *p* = 0.02) or heart diseases (OR 1.33, 95% CI 1.01–1.75, *p* = 0.04), were evidenced by the bivariate analysis (Table 2). In the multivariate regression (Table 2), only five independent predictors for the achievement of a sustained ROSC were identified: younger adults, witnessed cardiac arrest, prehospital ROSC, prehospital defibrillation, and comorbid diabetes mellitus.

### 3.3. The Impact of Crew Number and EMT-Paramedic Ratio on Patient Outcomes

After adjusting the abovementioned four independent predictors for sustained ROSC, the benefit on patient outcome (i.e., sustained ROSC and favorable neurologic status at discharge) from increasing the EMT-P ratio was identified (Table 3). Overall, an increase of 10% in the EMT-P ratio was, on average, associated with a 8% increased chance (aOR 1.08, 95% CI 1.02–1.13; *p* < 0.01) for sustained ROSC and a 12% increased chance (aOR, 1.12, 95% CI 1.01–1.26, *p* = 0.048) for favorable neurologic status at discharge. However, an increased number of total on-scene EMTs or EMT-Is was not linked to an increasing chance of any good patient outcome, whether sustained ROSC, survival to discharge, or favorable neurologic status at discharge (all *p* > 0.05). Furthermore, the increased number of the on-scene EMT-P was associated with an increasing chance of sustained ROSC, but not linked to those of survival to discharge, or favorable neurologic status at discharge.

## 4. Discussion

The number and proportion of EMTs on the scene for OHCA resuscitation has remained a modifiable factor linked to ROSC [13,14,15,16,17]. Until now, the influences of the EMT configuration on outcomes of OHCA patients remain controversial. Sam A. Warren et al. and Kajino et al. both indicated that the number of EMTs was associated with survival rate and even neurological outcome in patients with OHCA [13,14], whereas Hagiwara S. et al. and Eschmann NM et al. found opposite results [15,16]. In our cohort, the association of EMT number and patient outcomes, in terms of sustained ROSC, survival to discharge, and favorable neurologic status at discharge, were trivial after adjustment for all the independent predictors of sustained ROSC.

EMT-Ps are capable of administering more advanced management of OHCA patients [12] and are more confident in performing clinical activities and tasks compared with EMT-Is [18]. Therefore, the impact of EMT-Ps on the scene of resuscitation may not be equal to the impact of EMT-Is. J.T. Sun et al. demonstrated that a high EMT-P ratio (>50%) improved the survival rate of OHCA patients [17]. Our study also differentiated between levels of training certification among EMTs and observed similar findings. Furthermore, we found that an increase in the EMT-P ratio by 10% increased the chance of sustained ROSC by 8% and favorable neurologic status at discharge by 12%. In brief, adequate EMT-P ratios could be a crucial component of the EMS configuration.

There is a major explanation for our finding that a high EMT-P ratio resulted in improved outcomes among OHCA patients. Gold LS et al. indicated that the experience of the paramedic who performed procedures rather than the paramedic in charge was associated with the survival rate of OHCA patients [25]. By going through more training courses [10,11], EMT-Ps usually have more skill and confidence in performing OHCA resuscitation [18,25]. As one EMT-P leads the resuscitation team, the other EMT-P can execute other procedures, including intravenous catheter insertion, LMA, and provision of medication. Hence, an increasing ratio of EMT-Ps may increase the likelihood that an EMT-P will perform procedures and thereby improve the outcome of OHCA patients.

However, Bayley R et al. suggested that a two-EMT-P crew did not perform better than a single-EMT-P ambulance crew, either in interventions or completeness of resuscitation [26]. This conclusion is in opposition to our results and does not favor the inclusion of more EMT-P members. Bayley R et al. speculated that the indefinite leadership in two-EMT-P crews may contribute to the poor performance of resuscitation. Since this study only recruited 15 crews in each configuration and was conducted in a simulated situation, further real-world data are needed. Assuming the speculation of Bayley et al. is right, indefinite leadership could also be overcome by training courses or further education. Both teamwork and leadership training have been shown to improve team performance [27], as emphasized by American Heart Association guideline [28]. Therefore, indefinite leadership was not considered as an explanation in our study.

Among all EMTs, an EMT-P is authorized to perform intravenous epinephrine injection. In our cohort, a high proportion of epinephrine administration was found in the group with a high EMT-P ratio. The effect of prehospital epinephrine is still being debated, and a randomized trial involving 8014 patients comparing epinephrine with placebo showed a better rate of 30-day survival [29]. The reason that the increasing ratio of EMT-Ps did not improve the survival rate at discharge in this trial is uncertain. Our explanation is that intravenous catheter insertion on the scene may take too much time. Perkins et al. injected either epinephrine or placebo on the scene and found no difference in EMS time at the scene; however, we did not find a difference in the ratio of scene time, which may be modulated by better competency of EMT-Ps [18].

Several predictors of survival from OHCA have been identified in the literature, such as bystander CPR [1], electric shock therapy [1], ambulance response time [30], EMS scene time [31], EMS transport time [32], location of cardiac arrest [33], and adrenaline use [34]. Although only one predictor, electric shock therapy, was evidenced as an independent determinant of sustained ROSC in our cohort, two other variables, EMS response time ≤ 6 min and indoor location of arrest, were borderline-significant predictors under the multivariate analyses. We suspect that differences in the prehospital emergency systems between Taiwan and Western countries, such as city area, density of prehospital ALS stations and hospitals, and the number and proportion of EMT-Ps on the scene who are capable of CPR [35] and targeted temperature management [36], might also have resulted in the differences.

Several limitations should be considered when interpreting our findings. First, as a retrospective cohort study, this study had inherent problems with data collection. Although we had undergone quality control of CPR in the Tainan fire department by the national standard training program, it is difficult to calculate the individual difference in CPR operation between EMTs. Second, the different impact of on-scene numbers of EMT-Ps and EMT-Is on patient outcome was respectively studied, but the difference between EMT-1 and EMT-2 was not considered. Additionally, although we observed better outcomes when there was a high EMT-P ratio on the scene, the different dispatcher systems or EMS teams may have different outcomes. Third, compared to the previously established report in US and Canada [14], less EMT numbers in the ambulance, in which the majority (1,103, 81.3%) of cases was transported by 2 EMTs, was exhibited in our cohort. Therefore, the difficulty of external validation of our finding to other communities should be considered. Fourth, given that resuscitation skill has declined as time passed, exposure to OHCA treatment had become an important contributing factor. Previous studies suggested that recent exposure instead of career experience increased patient survival [37,38], which should be considered in future studies. Finally, the application of our study results should be tailored to local EMS practices since this study was conducted in an EMS system that adapts the policy of “resuscitation during transportation” for OHCAs [39,40].

## 5. Conclusions

Despite the neglected relationship between EMT number and patient outcomes, an increase in the EMT-P ratio resulted in an increased proportion of sustained ROSC and improved cerebral performance at discharge for nontraumatic OHCA adults transported by ALS teams. Accordingly, more EMT-P training programs are needed to augment the EMT-P number and thereby raise their on-scene ratio.

## Figures and Tables

**Figure 1 ijerph-17-01930-f001:**
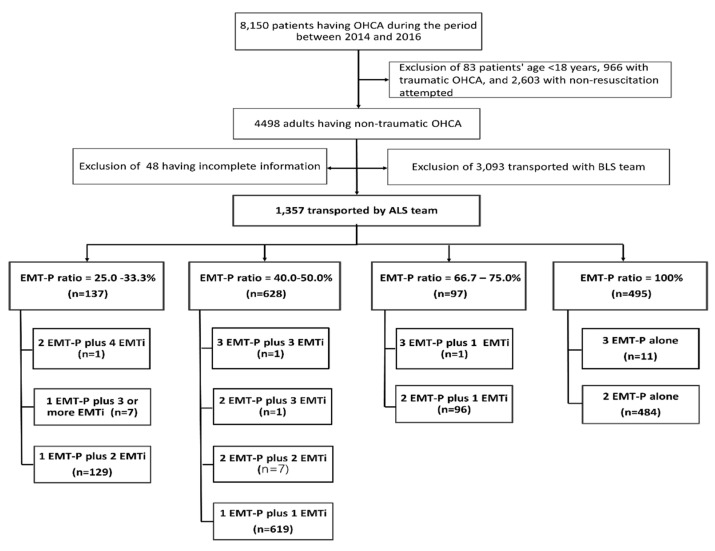
Patient selection flowchart. EMT = emergency medical technician; EMT-I = emergency medical technician-intermediate; EMT-P = emergency medical technician-paramedic; OHCA = out-of-hospital cardiac arrest. EMT-I included EMT-1 and EMT-2.

**Figure 2 ijerph-17-01930-f002:**
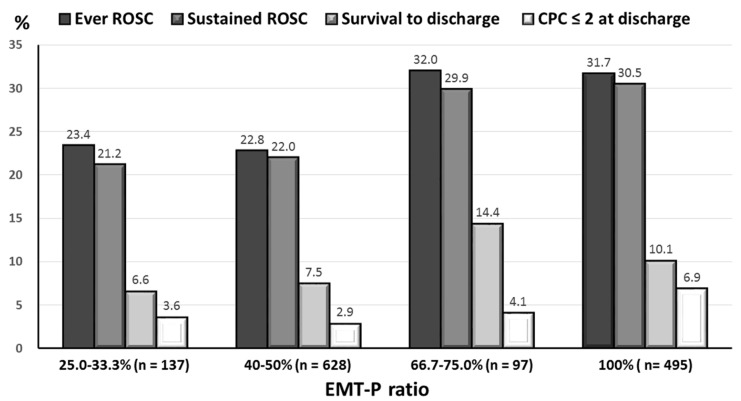
A positive EMT-P-related trend of the proportion of sustained ROSC), survival to discharge and CPC ≤ 2 at discharge. CPC = cerebral performance category; EMT-P = emergency medical technician-paramedic; ROSC = return of spontaneous circulation.

**Table 1 ijerph-17-01930-t001:** Demographic data and outcomes of enrolled out-of-hospital cardiac arrest patients stratified by the EMT-paramedic ratio.

Variables	Patient Number (%) in Varied Groups of EMT-Paramedic Ratio				*γ*	*p* Value
	25.0–33.3% n = 137	40.0–50.0% n = 628	66.7–75.0% n = 97	100.0% n = 495		
Younger adults (<65 years)	60 (43.8)	209 (33.3)	42 (43.3)	196 (39.6)	−0.40	0.60
Male	92 (67.2)	404 (64.3)	68 (70.1)	320 (64.6)	0	1.00
Witnessed cardiac arrest	66 (48.2)	315 (50.2)	50 (51.5)	230 (46.5)	−0.20	0.80
Bystander CPR	48 (35.0)	211 (33.6)	37 (38.1)	148 (29.9)	−0.40	0.60
Prehospital ROSC	14 (10.2)	22 (3.5)	7 (7.2)	48 (9.7)	−0.20	0.80
EMS time interval						
Response time ≤ 5 min	81 (59.1)	437 (69.6)	48 (49.5)	275 (55.6)	−0.60	0.40
Scene time ≤ 8 min	95 (69.3)	461 (73.4)	74 (96.3)	383 (77.4)	0.80	0.20
**Transport time ≤ 5 min**	**95 (69.3)**	**377 (60.0)**	**45 (46.4)**	**186 (37.6)**	**−1.00**	**0.01**
Indoor location of arrest	127 (92.7)	575 (91.6)	83 (85.6)	437 (88.3)	−0.80	0.20
Transport to medical centers	56 (40.9)	330 (52.5)	50 (51.5)	277 (56.0)	0.80	0.20
Prehospital treatment						
Laryngeal mask airway	114 (83.2)	514 (81.8)	83 (85.6)	423 (85.5)	0.60	0.40
Defibrillation	22 (16.1)	92 (14.6)	16 (16.2)	103 (20.8)	0.80	0.20
Intravenous epinephrine	9 (6.6)	24 (3.8)	28 (28.9)	127 (25.7)	0.60	0.40
Comorbidities						
Hypertension	48 (35.0)	221 (35.2)	30 (30.9)	184 (37.2)	0.40	0.60
Diabetes mellitus	37 (27.0)	186 (29.6)	21 (21.6)	136 (27.5)	0	1.00
Heart disease	39 (28.5)	149 (23.7)	19 (19.6)	132 (26.7)	−0.40	0.60
Chronic kidney disease	14 (10.2)	75 (11.9)	10 (10.3)	77 (15.6)	0.80	0.20
Neurological disease	26 (19.0)	100 (15.9)	19 (19.6)	67 (13.5)	−0.40	0.60
**Hemato-oncological disease**	**13 (9.5)**	**70 (11.1)**	**12 (12.4)**	**69 (13.9)**	**1.00**	**0.01**
COPD	10 (7.3)	37 (5.9)	14 (14.4)	34 (6.9)	0	1.00
Chronic liver disease	6 (4.4)	23 (3.7)	7 (7.2)	17 (3.4)	−0.40	0.60
Psychological disease	5 (3.6)	11 (1.8)	5 (5.2)	13 (2.6)	0	1.00
Outcomes						
Ever ROSC	32 (23.4)	143 (22.8)	31 (32.0)	157 (31.7)	0.60	0.40
**Sustained (≥ 2 h) ROSC**	**29 (21.2)**	**138 (22.0)**	**29 (29.9)**	**151 (30.5)**	**1.00**	**0.01**
Survive to discharge	9 (6.6)	47 (7.5)	14 (14.4)	50 (10.1)	0.80	0.20
Favorable neurologic status at discharge *	5 (3.6)	18 (2.9)	4 (4.1)	34 (6.9)	0.80	0.20

COPD = chronic obstructive pulmonary disease; CPR = cardiopulmonary resuscitation; EMS = emergency medical service; ROSC = return of spontaneous circulation. Boldface indicates statistical significance, i.e., a *p* value of <0.05. ***** Indicates the cerebral performance category level I and II at discharge.

**Table 2 ijerph-17-01930-t002:** Predictors of achievement of a sustained (≥2 h) return of spontaneous circulation.

Variables	Patient No (%) with ROSC		Univariate Analysis		Multivariate Analysis	
	Yes, n = 347	No, n = 1010	OR (95% CI)	*p* Value	Adjusted OR (95% CI)	*p* Value
Younger adults (<65 years)	165 (47.6)	342 (33.9)	**1.75 (1.39−2.27)**	**<0.01**	**1.76 (1.33−2.33)**	**<0.01**
Witnessed cardiac arrest	243 (70.0)	418 (41.4)	**3.31 (2.55−4.30)**	**<0.01**	**2.86 (2.16−3.78)**	**<0.01**
Bystander cardiopulmonary resuscitation	225 (64.8)	688 (68.1)	0.86 (0.67−1.12)	0.26	NS	NS
EMS response time ≤ 5 min	198 (57.1)	643 (63.7)	**0.76 (0.59−0.97)**	**0.03**	NS	NS
Prehospital ROSC	75 (21.6)	16 (1.6)	**17.13 (9.82−29.87)**	**<0.01**	**12.48 (7.03−22.16)**	**<0.01**
Indoor location of arrest	298 (85.9)	924 (91.5)	**0.57 (0.39−0.82)**	**<0.01**	NS	NS
Prehospital management						
Defibrillation	94 (27.1)	139 (13.8)	**2.33 (1.73−3.13)**	**<0.01**	**1.45 (1.03−2.04)**	**0.03**
Intravenous epinephrine	60 (17.3)	128 (12.7)	**1.44 (1.03−2.01)**	**0.03**	NS	NS
Comorbidities						
Diabetes mellitus	114 (32.9)	266 (26.3)	**1.37 (1.05−1.78)**	**0.02**	**1.48 (1.11−1.99)**	**<0.01**
Heart disease	101 (29.1)	238 (23.6)	**1.33 (1.01−1.75)**	**0.04**	NS	NS
Chronic kidney disease	54 (15.6)	122 (12.1)	1.34 (0.95−1.90)	0.09	NS	NS

CI = confidence interval; EMS = emergency medical service; NS = not significant (after processing the backward multivariate regression); OR = odds ratio; ROSC = return of spontaneous circulation. Boldface indicates statistical significance, i.e., a *p* value of <0.05.

**Table 3 ijerph-17-01930-t003:** Impacts of the crew number and EMT-paramedic ratio on patient outcomes by multivariate analysis.

Variables	Sustained ROSC		Survival at Discharge		CPC ≤ 2 at Discharge	
	Adjusted OR * (95% CI)	*p* Value	Adjusted OR * (95% CI)	*p* Value	Adjusted OR * (95% CI)	*p* Value
**EMT number (averagely increased one person)**						
Total	1.02 (0.76−1.37)	0.89	1.03 (0.68−1.57)	0.89	1.19 (0.68−2.09)	0.55
EMT-intermediate	1.26 (0.99−1.55)	0.06	1.13 (0.84−1.52)	0.44	1.44 (0.94−2.20)	0.09
EMT-paramedic	**1.49 (1.14−1.93)**	**0.003**	1.23 (0.82−1.84)	0.32	1.76 (0.98−3.15)	0.06
EMT-paramedic ratio (averagely increased the ratio of 10%)	**1.08 (1.02−1.13)**	**<0.01**	1.03 (0.95−1.11)	0.54	**1.12 (1.01−1.26)**	**0.048**

CI = confidence interval; CPC = cerebral performance category; EMT = emergency medical technician; OR: odds ratio; ROSC = return of spontaneous circulation. Boldface indicates statistical significance, i.e., a *p* value of <0.05. * Adjusted for five independent predictors: younger adults, witnessed cardiac arrest, prehospital ROSC, prehospital defibrillation, and comorbid diabetes mellitus. Boldface indicates statistical significance, i.e., a *p* value of <0.05.

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
