# Peer review of "Impacts of Emergency Medical Technician Configurations on Outcomes of Patients with Out-Of-Hospital Cardiac Arrest"

_ijerph, 2020, doi:10.3390/ijerph17061930_

Round 1
Reviewer 1 Report
This paper is well written and described the manpower of EMT in OHCA.
Interestingly, the survival rate is similar but the initial ROSC rate is different.
Author Response
Response: Many thanks for your positive review. For non-traumatic OHCA patients with sustained ROCS in the ED, we believed that numerous covariates were linked to their survival at discharge. The primary and secondary outcome assessed in the present study was consistent with those in our previous study (Ref No. 22). Please refer to Line 127-130 on Page 3.
Reviewer 2 Report
Thank you very much for the opportunity to review this manuscript.
The authors conducted a retrospective observational trial, aimed to investigate the impact of a higher proportion of EMT-P within the EMS response teams on ROSC and survival after nontraumatic out of hospital cardiac arrest.
The authors found out that a higher proportion of EMT-P at the scene leads to a significantly higher rate of sustained ROSC and to a trend in improved survival and neurological outcome.
Overall, the manuscript seems well written and the key points are pointed out understandably.
I appreciate the approach of sending out highly trained paramedics to the scene in case of emergency. I therefore consider this work to be clinically relevant.
General Comments/questions:
- the statement that higher education leads to an improved outcome is on the one hand known and is probably, as stated by the authors, due to the higher rate of ALS applications than BLS applications, (trend to higher proportion of defibrillation and administration of epinephrine within groups of more than 50% EMT-P) and (as also pointed out) gained experience and confidence.
- In Figure 1, the authors described an exclusion of 3093 patients who were transported by BLS Team, in the result section you are describing 2603 patients without CPR attempt. Does the BLS Teams omitted CPR on transport? Please comment
- Was the alarm code in all cases “Resuscitation”?
- Did the dispatcher know the EMS configuration?
- Due to the “scoop and run” principle, the ongoing CPR efforts (ALS) during transport represent an important period. Was the ambulance driver always an EMT-I?
- How many patients achieved ROSC on transport?
- Overall the authors conclude that a higher proportion of EMT-P at the scene increases outcome. Please comment the “missing” differences in outcome between the 75% 100% group.
- Can you provide data about endotracheal intubation rates?
Specific Comments:
- in figure1, one patient is wrongly pointed out to be in a ratio of 50% (2 EMT-P, 3 EMTi)
- I would recommend to change the “eldery patient” to “younger patients” to point out a positive correlation.
- Line 192: Please add all CI’s or delete this one
Author Response
Thank you very much for the opportunity to review this manuscript.
The authors conducted a retrospective observational trial, aimed to investigate the impact of a higher proportion of EMT-P within the EMS response teams on ROSC and survival after nontraumatic out of hospital cardiac arrest. The authors found out that a higher proportion of EMT-P at the scene leads to a significantly higher rate of sustained ROSC and to a trend in improved survival and neurological outcome. Overall, the manuscript seems well written and the key points are pointed out understandably. I appreciate the approach of sending out highly trained paramedics to the scene in case of emergency. I therefore consider this work to be clinically relevant.
Response: Many thanks for review and your positive opinion about our work.
General Comments/questions:
the statement that higher education leads to an improved outcome is on the one hand known and is probably, as stated by the authors, due to the higher rate of ALS applications than BLS applications, (trend to higher proportion of defibrillation and administration of epinephrine within groups of more than 50% EMT-P) and (as also pointed out) gained experience and confidence.
- In Figure 1, the authors described an exclusion of 3093 patients who were transported by BLS Team, in the result section you are describing 2603 patients without CPR attempt. Does the BLS Teams omitted CPR on transport? Please comment.
Response: Thanks for your question. In Figure 1, 2603 patients with non-resuscitation were excluded because of do-not-resuscitation (DNR) or other requirement by family (Line155-156, Page 4). Because effective ACLS was not completely performed in BLS team, only OHCA patients transported by the ALS team was eligible in our work (Line 76-78, Page 2).
- Was the alarm code in all cases “Resuscitation”?
Response: Yes.
- Did the dispatcher know the EMS configuration?
Response: Generally, the dispatchers did not know the exact EMS configuration. The dispatchers would dispatch ALS teams should events of cardiac arrest be recognized in the emergency calls. Otherwise, the dispatchers usually dispatch the nearest teams to the scene.
- Due to the “scoop and run” principle, the ongoing CPR efforts (ALS) during transport represent an important period. Was the ambulance driver always an EMT-I?
Response: The ambulance driver was not necessarily always EMT-I. However, the most experienced EMTs are usually responsible of taking care the patients during transportation.
- How many patients achieved ROSC on transport?
Response: Although the detailed information about the number of patients achieved ROSC during the EMT transport was not available in our work, a new covariate of "prehospital ROSC" was added in our new work. Accordingly, all Tables were recalculated and revised. More importantly, similar conclusion that an "increase in the EMT-P ratio by 10% increased the chance of sustained ROSC by 8% and favorable neurologic status at discharge by 12%" was evidenced in our new work. The related sentences were revised throughout the manuscript (Line 35-38. Page 1; Line 189-190, 195, Page 6; Line200-202 Page 7; and Line 229-230, Page 9).
- Overall the authors conclude that a higher proportion of EMT-P at the scene increases outcome. Please comment the “missing” differences in outcome between the 75% and 100% group.
Response: Thanks for your suggestion. A great diversity of EMT number was noticed worldwide. For example, the median number of EMTs is 5 persons in the cohort of the Canada and the United States, indicated by Dr. Warren (Ref. No. 14). However, the numbers of EMTs in our cohort was consistent with those in Japan cohort (Ref. No. 13; median, 2 persons). Because of the limited EMT numbers, to overcome the proportion gap of the EMT-P ratio, the logistic regression method (Line140-142, Page 4) was adapted in our work to identify the value of increased chance of sustained ROSC or favorable neurologic status at discharge, with regard to an increasing proportion of the EMT-P ratio. Our finding was listed in Line 200-203, Page 7. Our consideration about the difficult of external validation was added in the paragraph of "limitation"(Line 275-278, Page 10)
- Can you provide data about endotracheal intubation rates?
Response: During the study period, the performance of endotracheal intubation was illegal by the EMT-P in study area. Thus, only information about Laryngeal mask airway was available in our work.
Specific Comments:
- in figure1, one patient is wrongly pointed out to be in a ratio of 50% (2 EMT-P, 3 EMTi)
Response: The error in the grouping name in Figure 1 and Table 1 was revised. This patient was correctly categories in the ratio group of 40.0-50.0 %.
- I would recommend to change the “elderly patient” to “younger patients” to point out a positive correlation.
Response: Thanks for your suggestion. the term of "younger adults" was used in Table 1 and throughout the revised manuscript, in place of the term "elderly patients".
Line 192: Please add all CI’s or delete this one
Response: This error was corrected (Line 196, Page 6).
Reviewer 3 Report
Education and quality management of EMTs are important issues in public health.
1) This article covers and tried to answer one of the most critical problems of prehospital resuscitation.
However, numbers of EMTs on-scene are only 2 or 3 for most cases. Due to small numbers of on-scene EMTs, your results cannot fully answer for the proportion issues of EMT-P.
I think you should focus on the specialized EMT-P only team for prehospital resuscitation outcomes. Groups can be divided with EMT-i only, mixed (EMT-P and EMT-i ), and EMT-P only.
Also, Total numbers of on-scene EMTs are considered for outcome analysis.
2) The criteria for EMT-P dispatch is not clear. There may be a selection bias. For example, sometimes experienced rescuers are insufficient during the night time or holidays.
3) If you stick to the proportion issue, the total EMT-p number should be adjusted for analysis.
Author Response
Education and quality management of EMTs are important issues in public health.
- This article covers and tried to answer one of the most critical problems of prehospital resuscitation. However, numbers of EMTs on-scene are only 2 or 3 for most cases. Due to small numbers of on-scene EMTs, your results cannot fully answer for the proportion issues of EMT-P.
I think you should focus on the specialized EMT-P only team for prehospital resuscitation outcomes. Groups can be divided with EMT-i only, mixed (EMT-P and EMT-i ), and EMT-P only.
Also, Total numbers of on-scene EMTs are considered for outcome analysis.
Response. Many thanks for your opinion and suggestions. We agreed that less numbers of EMTs in the study city, compared to other area. For example, the median number of EMTs is 5 persons in the cohort of the Canada and the United States, indicated by Dr. Warren (Ref. No. 14). But, the numbers of EMTs in our cohort was similar to those in Japan cohort (median, 2 persons), as description in Ref. No.13. Your consideration about the difficult of external validation was reasonable and was added in the paragraph of "limitation"(Line 275-278, Page 10).
We respectfully appreciate your suggestion about the group categorization of EMT configurations. Superior to previous reports dealing with EMT configurations (Ref. No. 15 and 17), the statistic method used in our work to identify the value of increased chance of sustained ROSC or favorable neurologic status at discharge, with regard to an increase of 10% in the EMT-P ratio, when respectively adjusting all the independent predictors of sustained ROSC. Our finding was listed in Line 200-203, Page 7.
- The criteria for EMT-P dispatch is not clear. There may be a selection bias. For example, sometimes experienced rescuers are insufficient during the night time or holidays.
Response: Many thanks for your opinion. The detailed information of EMT dispatch was inverted in the revised manuscript (Line 88-91, Page 2). Because EMT-P dispatch is performed all the day, the selection bias should be neglected.
- If you stick to the proportion issue, the total EMT-P number should be adjusted for analysis.
Response: Thanks for your suggestion. In addition to EMT configurations, the impact of the EMT number on patient outcomes was also studied in our study design (Line139-141, Page 3-4). Our finding was listed in Line 203-205 (Page 7) and Table 3.
Round 2
Reviewer 2 Report
Thank you for the revision of the manuscript.
The authors claryfied all issues to my satisfaction.
Author Response
Many thanks your review again.
Reviewer 3 Report
major
#1. Your data according to total Numbers of ALS team.
6 EMTs only 2 cases
5 EMTs only 1 cases
4 EMTs 15 cases
3 EMT 236 cases
2 EMT 484 cases
Because of a few cases in 6,5 and 4 EMTs, your results cannot fully answer for the numbers of total EMT or EMT-P issues.
I also think the more, the better for EMT-P issues, your results didn't cover many situations. Actually, I think your data only explains outcome differences between 1 EMTP vs. 2EMTP. (Table)
| Gr 1 | Gr 2 | Gr 3 | Gr4 |
|
1EMTP + 2 EMTi 94% |
1 EMTP+1 EMTi 99% |
2 EMTP+ 1 EMTi 98% | 2EMTp only 98% |
|
Gr 1 vs G2 no difference for outcomes (chi square test) |
Gr 1 vs G2 no difference for outcomes (chi square test) |
Gr 3 vs G4 no difference for outcomes (chi square test) | Gr 3 vs G4 no difference for outcomes (chi square test) |
I recommend you consult with a statistician for interpreting the results of logistic regression results. EMT-P proportion and outcome plot can be helpful for understanding.
minor
Line 203 However, increased proportion of EMTs was not linked -> I think you mean not “proportion” but “numbers”.
Author Response
Major
Your data according to total Numbers of ALS team.
6 EMTs only 2 cases
5 EMTs only 1 cases
4 EMTs 15 cases
3 EMT 236 cases
2 EMT 484 cases
Because of a few cases in 6,5 and 4 EMTs, your results cannot fully answer for the numbers of total EMT or EMT-P issues.
I also think the more, the better for EMT-P issues, your results didn't cover many situations. Actually, I think your data only explains outcome differences between 1 EMTP vs. 2EMTP. (Table)
|
Gr 1 |
Gr 2 |
Gr 3 |
Gr4 |
|
1EMTP + 2 EMTi 94% |
1 EMTP+1 EMTi 99% |
2 EMTP+ 1 EMTi 98% |
2EMTp only 98% |
|
Gr 1 vs G2 no difference for outcomes (chi square test) |
Gr 1 vs G2 no difference for outcomes (chi square test) |
Gr 3 vs G4 no difference for outcomes (chi square test) |
Gr 3 vs G4 no difference for outcomes (chi square test) |
I recommend you consult with a statistician for interpreting the results of logistic regression results. EMT-P proportion and outcome plot can be helpful for understanding.
Response: Many thanks for your opinions. In according to your considerations, our new work was respectively listed in the Supplemental Table and the revised Table 3. After communications with a statistician, there were three leading reasons to support our statistic method using the logistic regression model to investigate the impact of the EMT-P ratio and the EMT number on patient outcomes
First, based on your above opinions, patients delivered by ≥ 3 numbers of on-scene EMT-Ps were excluded. Although these cases only accounted for few proportion (13/1,357, 1.0 %) of the overall cohort, the selection bias should be considered because these excluded patients were delivered by the extreme high number of EMT-Ps, which is indicated of the measure (independent) variable in our study design.
Second, the comparisons of the varied predictors between Group 1, 2, 3, and 4 was disclosed in the Supplemental Table. In this Table, the similarity of patient outcomes between the Group 1 and 2 was disclosed, but the dissimilar proportion of numerous variables, such as younger adults, the EMS response time ≤ 5 min, and prehospital ROSC, between two groups, was demonstrated. More importantly, there variables were associated with sustained ROSC by the Chi-square or multivariate analysis, as shown in Table 2. Therefore, by the way, it was unreasonable to conclude the neglected impact of EMT-I numbers on patient outcome.
Third, to our knowledge, the logistic regression model is used when the dependent variable(target) is categorical and the independent variables can each be a binary variable (two classes, coded by an indicator variable) or a continuous variable (any real value). After the analyses, the probability of dependent variable 1 is relatively several times higher than that of dependent variable 0, when the independent variable is increased by one unit. Accordingly, our work is enough academic to study the impact of the varied EMT configuration on patient outcomes, using the Group 1 and 2. Furthermore, in our new work, impacts of the on-scene numbers of EMT-Ps and EMT-Is on patient outcomes was additionally analyzed by the logistic regression model, respectively (as shown in the revised Table 3). In conformity with our previous result, the beneficial influence of the increased EMT-P ratio remained as the most crucial finding in our work.
Minor
Line 203 However, increased proportion of EMTs was not linked -> I think you mean not “proportion” but “numbers”.
Response: As the above statement, this sentence and another description about the new Table 3 were revised (Line 203-207, Page 7).
